# Morindone from *Morinda citrifolia* as a potential antiproliferative agent against colorectal cancer cell lines

Cheok Wui Chee[1], Nor Hisam Zamakshshari[2], Vannajan Sanghiran Lee[3,4,5]*, Iskandar Abdullah[3,4], Rozana Othman[4,6,7], Yean Kee Lee[3,4], Najihah Mohd Hashim[4,6,7]*, Nurshamimi Nor Rashid[1,4,6]*

1 Department of Molecular Medicine, Faculty of Medicine, Universiti Malaya, Kuala Lumpur, Malaysia, 2 Department of Chemistry, Faculty of Resources Science, Universiti Malaysia Sarawak, Sarawak, Malaysia, 3 Department of Chemistry, Faculty of Science, Universiti Malaya, Kuala Lumpur, Malaysia, 4 Drug Design and Development Research Group, Universiti Malaya, Kuala Lumpur, Malaysia, 5 Centre of Theoretical and Computational Physics, Faculty of Science, Universiti Malaya, Kuala Lumpur, Malaysia, 6 Centre for Natural Products Research and Drug Discovery, Universiti Malaya, Kuala Lumpur, Malaysia, 7 Faculty of Pharmacy, Universiti Malaya, Kuala Lumpur, Malaysia

* nurshamimi@um.edu.my (NNR); najihahmh@um.edu.my (NMH); vannajan@um.edu.my (VSL)

**Data Availability Statement:** All relevant data are within the paper and its Supporting information files.

## Abstract

There is an increasing demand in developing new, effective, and affordable anti-cancer against colon and rectal. In this study, our aim is to identify the potential anthraquinone compounds from the root bark of *Morinda citrifolia* to be tested *in vitro* against colorectal cancer cell lines. Eight potential anthraquinone compounds were successfully isolated, purified and tested for both *in-silico* and *in-vitro* analyses. Based on the *in-silico* prediction, two anthraquinones, morindone and rubiadin, exhibit a comparable binding affinity towards multitargets of β-catenin, MDM2-p53 and KRAS. Subsequently, we constructed a 2D interaction analysis based on the above results and it suggests that the predicted anthraquinones from *Morinda citrifolia* offer an attractive starting point for potential antiproliferative agents against colorectal cancer. *In vitro* analyses further indicated that morindone and damnacanthal have significant cytotoxicity effect and selectivity activity against colorectal cancer cell lines.

## Introduction

Colorectal cancer (CRC) as a malignant cancer affecting both male and female, is ranked the third most common cancer worldwide and second most frequent cancer in Malaysia [1]. Significant associations between dietary factors and CRC risk have been determined, in addition to smoking and alcohol intake [2]. Regular consumption of fruits and vegetables was demonstrated effective in reducing CRC risk as polyphenolic compounds in plants contribute to decreasing cell adhesion process, migration, and tumour angiogenesis [3]. Standard chemotherapy regimens in treating CRC patients accommodates the use of cancer drug particularly 5-fluorouracil (5-FU) and doxorubicin hydrochloride (DOX) that function by inhibiting DNA synthesis [4]. Despite higher survival rate in patients, the adverse toxicity risk associated with

**Funding:** NNR, IIRG003C-2019 IA,IIRG003A-2019 Impact Oriented Interdisciplinary Research Grant Program. The funders had no role in study design, data collection and analysis, decision to publish, and or preparation of the manuscript.

**Competing interests:** The authors have declared that no competing interests exist.

these chemotherapy drugs need to be taken into account [5]. Therefore, the search for effective phytochemical compounds as antiproliferative agent continues.

Anthraquinone, an aromatic compound with a 9, 10-dioxoanthracene core can be found abundantly in several plants such as rhubarb root, aloe vera, morinda and senna leaf [6]. It is reported to display pharmacological properties including anti-inflammatory, antioxidant, anti-microbial and anticancer [7, 8] has reported that anthraquinone compounds obtained from *Morinda citrifolia* exhibited promising *in vitro* antitumor activity and selective against CRC cells. Following this, eight *Morinda citrifolia* isolated anthraquinone compounds were evaluated for cytotoxicity activity against CRC cell lines. The combination therapy effect of anthraquinone compounds, 5-FU and DOX was also explored.

CRC is associated with a series of genetic alteration involving various pathway such as Wnt signalling pathway, Ras signaling pathway and p53 mediated apoptosis pathway. Being the crucial players of these pathway, β-catenin gene is constantly degraded by adenomatous polyposis coli (APC) [9] while p53 and Kirsten rat sarcoma virus (KRAS) genes are regularly found mutated in CRC [10]. Moreover, patients harboring KRAS mutation generally acquire resistance to anti-epidermal growth factor receptor (EGFR) therapy, conferring poor survival [11]. [12] work in 2012 has reported promising combination therapeutic strategy in CRC by down regulating both β-catenin and KRAS simultaneously. Meanwhile, study by [13] showed that the autoregulatory negative feedback loop involving regulation of p53 by Murine Double Minute 2 (MDM2) could be a novel target therapy in cancer treatment as these two genes regulate each other mutually through an autoregulatory negative feedback loop. Plenty of evidence suggests the competent anticancer effect of anthraquinone against various genes and human cancers, including our target genes: β-catenin [14], MDM2-p53 [15–17], and KRAS [18, 19]. Given the lack of research elucidating the effect of anthraquinone in CRC, this study aim to discover novel potential inhibitors based on anthraquinones from *Morinda citrifolia* through the deployment of ligand-based protein docking for β-catenin, MDM2-p53 and KRAS. The discovery of new anticancer drugs by aiming the protein products [20] and aided using computational techniques [21] have been reported. Herewith, both *in silico* and *in vitro* work was proposed to investigate the correlation between structure conformation of anthraquinone and its cytotoxicity in CRC.

## Materials and methods

### 2.1 Chemicals

All chemicals and solvents were purchased from Merck (Germany), Sigma-Aldrich (St Louis, MO, USA) and Thermo Fisher Scientific (Waltham, MA, USA). MTT Cell Count Kit for cell cytotoxicity assay was purchased from Nacalai Tesque (Kyoto, Japan). 5-fluorouracil (5-FU) and doxorubicin hydrochloride (DOX) were obtained from AoBo Bio-Pharmaceutical Technology (Shanghai, China) and Santa Cruz Biotechnology (CA, USA), respectively.

### 2.2 Plant collection, extraction, and isolation of anthraquinone

The root bark of *Morinda citrifolia* was collected from Negeri Sembilan, Malaysia and authenticated by Professor Dr. Rusea Go from the Biology Department, Faculty of Science, Universiti Putra Malaysia, Malaysia. A voucher specimen (RG6085) was deposited in the Herbarium of Biology Department, Faculty of Science, Universiti Putra Malaysia, Malaysia. The study complies with national regulations and all necessary permits were obtained for the same. The collected plant sample was dried under open air and ground into fine powder. The powdered root bark of *Morinda citrifolia* was macerated three times in four different solvents: hexane, chloroform, ethyl acetate and methanol for 72 hours. The macerated sample was filtered and

evaporated under reduced pressure to obtain dry extracts of methanol, ethyl acetate, chloroform, and hexane. Each extract was fractionated using vacuum column chromatography and resulted in 20 to 30 fractions. Then, each fraction underwent purification using two purification method which are gravity column chromatography and preparative thin layer chromatography. The isolation and purification of all extracts resulted in 8 anthraquinones as below:

**Nordamnacanthal (1)**. Yellow needle crystal; m.p: 219-222°C (literature 218-220°C, [22]) IR $v_{max}$ cm$^{-1}$: 2928, 1734, 1649, 1356, 1274; UV (EtOH) λmax 290 and 246nm; EI-MS m/z: 268, 240, 212, 184, 128, 69;$^1$H and $^{13}$C-NMR spectra are in consistent with literature [23].

**Damnacanthal (2)**. Yellow needle crystal; m.p: 208-210°C (literature 211-212°C, [24]) IR $v_{max}$ cm$^{-1}$: 2927, 1664, 1441, 1282 and 1116; UV (EtOH) λmax nm: 392 and 258; EI-MS m/z:282, 254, 225, 208, 139, 76, 63, 50; $^1$H and $^{13}$C- NMR spectra are in consistent with literature [23].

**1,3,5-trihydoxy-2-methoxy-6-methyl anthraquinone (3)**. Yellow powder; m.p: 235-239°C (literature 241°C, [22]) IR $v_{max}$ cm$^{-1}$: 3405, 2931, 1737, 1406, 1259 and 1161; UV (EtOH) λmax nm: 412 and 302; EI-MS m/z: 300,282,257,135,115,77,63,51. $^1$H and $^{13}$C- NMR spectra are in consistent with literature [22].

**Morindone (4)**. Orange-red needle crystal; m.p: 248-250°C (literature 250-251°C, [25]) IR $v_{max}$ cm$^{-1}$: 2919, 2854, 1597, 1273 and 1072; UV (EtOH) λmax nm: 422 and 269; EI-MS m/z: 270,242, 139, 77, 69, 51.$^1$H and $^{13}$C- NMR spectra are in consistent with literature [23].

**Sorendidiol (5)**. Yellow needle crystal; m.p: 286-288°C (literature 287-288°C, [26]) IR $v_{max}$ cm$^{-1}$:3295, 2930, 1740, 1561, 1440, 1294 and 1108; UV (EtOH) λmax nm: 412 and 320; EI-MS m/z: 254, 225, 197, 139, 115; $^1$H and $^{13}$C- NMR spectra are in consistent with literature [23].

**Rubiadin (6)**. Yellow needle crystal; m.p: 288-290°C (literature 290-291°C, [27]) IR $v_{max}$ cm$^{-1}$: 3383, 2918, 1733, 1580, 1428, 1306 and 1106 UV (EtOH) λmax nm: 419, 324 and 225; EI-MS m/z: 254, 226, 197, 152, 115, 76; $^1$H and $^{13}$C-NMR spectra are in consistent with literature [23].

**Damnacanthol (7)**. Yellow needle crystal; m.p: 156-158°C (literature 157°C, [24]) IR $v_{max}$ cm$^{-1}$: 3066, 2925, 1653, 1565, 1444, 1339 and 1258; UV (EtOH) λmax nm: 427, 282 and 215; EI-MS m/z:284, 269, 251, 237, 181, 152, 139, 76; $^1$H and $^{13}$C- NMR spectra are in consistent with literature [24].

**Lucidin-ω-methylether (8)**. Orange-Yellow powder; m.p: 170-173°C (literature 170°C, [28]) IR $v_{max}$ cm$^{-1}$: 3142, 2924, 1672, 1573, 1367, 1332 and 1273; UV(EtOH) λmax nm: 419, 325 and 226; EI-MS m/z: 284, 252, 196, 168; $^1$H and $^{13}$C- NMR spectra are in consistent with literature [28].

## 2.3 *In silico* investigation toward multi-protein targets in colorectal cancer

In order to elucidate the potential mechanism by which the active compounds induce the cytotoxic activity, molecular docking was performed to position the compounds into the active site of multiple targets; β-catenin (Protein Data Bank (PDB) ID: 1JDH), MDM2-p53 (PDB ID: 4HG7) and KRAS (PDB ID: 5OCT), where the binding sites are as the following:

1. β-catenin binding sites: Site A of β-catenin contains the key important amino acid Lys312 (Binding center x, y, z: -6.30, 1.85, 50.08) while Site B contains amino acid Lys435 (Binding center x, y, z: -1.56, 10.65, 21.00) which is important in the interaction between the protein and the TCF4 protein [29].

2. Crystal Structure of an MDM2/Nutlin-3a complex was used. MDM2-p53 binding site was selected from the ligand binding center at the position x, y, z: -24.55, 6.51, -13.83.

3. Small molecule inhibitor of Ras-effector protein interactions in X-ray structures was investigated and the ligand binding center at the position x, y, z: 64.00, 111.00, 1.81 was used.

All protein structures were prepared for docking using the protein preparation in Chimera software with default protocol for PDB2PQR and Dock Prep. Protonation state was assigned using PROPKA at pH 7.0 and gasteiger charges were assigned for protein [30]. Eight anthraquinone structures were downloaded from PubChem database except compounds 3, 5, 8 were modified from damnacanthol. All structures were optimized using PM6 level using Gaussian software package. Molecular docking was performed with local search algorithm using Autodock Vina in PyRx virtual screening software (http://pyrx.scripps.edu/). The prediction incorporated the target conformation as a rigid unit meanwhile allowing the ligands to be flexible and adaptable to the target. The amino acid binding sites were selected at the binding sites from the information above. Grid box was set at 20 x 20 x 20 $Å^3$ with the default grid spacing of 0.375. The three-dimensional (3D) structures were visualized by YASARA (http://www.yasara.org/index.html) and two-dimensional (2D) interaction was analyzed by BIOVIA Discovery Studio software (Dassault Systèmes BIOVIA, Discovery Studio Modeling Environment, Release 2018, San Diego: Dassault Systèmes, 2016). Different conformations for each ligand and the lowest binding affinity conformations were predicted.

## 2.4 Cell lines

Three human colorectal cancer cell lines: HT29 cells (colorectal adenocarcinoma with p53 and APC mutation), LS174T cells (colorectal adenocarcinoma with KRAS mutation) and HCT116 cells (colorectal carcinoma with KRAS mutation) as well as normal colon, CCD841 CoN cells were obtained from American Type Cell Culture (Manassas, VA, USA). All cell lines were cultured in an incubator under the condition of 37°C temperature, humidified 5% $CO_2$ air atmosphere. Cells were maintained in Dulbecco's Modified Eagle Medium (DMEM) (Corning, NY, USA), supplemented with 10% Fetal Bovine Serum (FBS) (Sigma-Aldrich, St Louis, MO, USA), 25mM HEPES (Sangon Biotech, Shanghai, China) and 1% Penicillin/Streptomycin (Merck, Germany).

## 2.5 Cell cytotoxicity assay

HCT116 wells were seeded at 7,500 cells/well while other cells were seeded at 30,000 cells/well in 96 well plates and allowed for 24 hours incubation. Cells were then treated with anthraquinone compounds at series of 2-fold dilution concentrations (50μM, 25μM, 12.5μM, 6.25μM, 3.12μM, 1.56μM, 0.78μM, 0.39μM) for 72 hours. 5-fluorouracil and DOX were used as positive controls in the treatment. Cell cytotoxicity was measured using the MTT Cell Count kit according to the manufacturer's protocol. Assay was performed in triplicates and repeated in three individual experiments. The absorbance (A) was measured with microplate reader at 570nm wavelength and normalized to control. The half-maximal inhibitory concentration ($IC_{50}$) value was determined using GraphPad Prism 8 software (GraphPad Software, Inc., CA, USA). Cell viability was calculated as follows:

$$\text{Cell viability } (\%) = \frac{A_{\text{untreated}} - A_{\text{blank}}}{A_{\text{treated}} - A_{\text{blank}}} \text{ x } 100$$

### 2.6 Selectivity index

Selective index (SI) value was measured to figure how much each anthraquinone compound is selective towards CRC cells in comparison with normal cell using the formula below:

$$\text{Selectivity Index} = \frac{\text{IC}_{50} \text{ in normal cells}}{\text{IC}_{50} \text{ in cancer cells}}$$

### 2.7 Drug combination assay

To identify the compound and drug combination is synergistic or antagonistic, the combination index of each pair at 50% inhibition was calculated. Only damnacanthal and morindone were selected for further study after compound screening. HCT116 and HT29 cells were seeded at 7,500 and 30, 000 cells/well respectively in 96 well plate. After 24 hours incubation for attachment, cells were treated in a multiple drug treatment using the method of constant-ratio drug combination proposed by Chou & Talalay in 1984 [31]. The combination treatment included damnacanthal and morindone at 2-fold concentration range of 0.39μM to 50μM along with 5-FU and DOX at 2-fold concentration range of 0.08μM to 10μM. MTT was carried out with absorbance read at 570nm after 72 hours of treatment. Assay was performed in triplicates and repeated in three individual experiments. Combination index was calculated using CompuSyn (ComboSyn, Inc., NJ, USA). Combination index lesser than 1 is considered as synergism effect, equal to 1 is addictive, and more than 1 is antagonism.

### 2.8 Statistical analysis

Statistical analysis was performed using GraphPad Prism 8 software. All data were presented as mean ± standard deviation (SD) of triplicates in three independent experiments. P value <0.05 is considered statistically significant.

## Results and discussion

### Isolation and purification of *Morinda citrifolia*

The root of *Morinda citrifolia* (3.0 kg) was extracted using macerated method in methanol (60 g), ethyl acetate (54 g), chloroform (38 g) and hexane (31 g). These crude extracts were then subjected under column and preparative thin layer chromatography which resulted in eight types of anthraquinone. Four out of eight anthraquinones were successfully isolated from the hexane extract namely nordamnacanthal (1) (27mg), damnacanthal (2) (3mg), 1,3,5-dihydoxy-2-methoxy-6-methyl anthraquinone (3) (3mg), and morindone (4) (15mg). Meanwhile, other anthraquinones were successfully isolated from the semi polar solvent namely sorendidiol (5) (21mg), rubiadin (6) (9.8mg), damnacanthol (7) (69.9mg) and lucidin-ω-methylether (8) (74.7mg). The $^1$H and $^{13}$C nuclear magnetic resonance (NMR) spectroscopic data (Supporting information) for compounds 1, 2, 4, 5, 6 were in consistence with literature [23]. While compounds 3, 7 and 8 were in consistence with literature [22, 24] and [28], respectively. The chemical structures of anthraquinone compounds 1–8 were illustrated in Fig 1.

### Docking of anthraquinones and binding interaction towards multitargets in colorectal cancer

Molecular docking was performed to determine the interaction between 8 isolated anthraquinones and the targeted protein receptors in cancer cell to explore the relationship between chemical structure and bioactivity. Results from molecular docking was validated by using the

**Fig 1. Chemical structure of anthraquinone compounds 1–8.** (1) nordamnacanthal, (2) damnacanthal, (3) 1,3,5-dihydoxy-2-methoxy-6-methyl anthraquinone, (4) morindone, (5) sorendidiol, (6) rubiadin, (7) damnacanthol, and (8) lucidin-ω-methylether.

bound inhibitors and redocked to its original position in the X-ray crystal structure. This binding model derived from a molecular docking was used to understand the anticancer mechanism at molecular level towards three targets, β-catenin, MDM2-p53, and KRAS. From our previous study [32], the Wnt–β-catenin pathway appears to be deregulated in most cancer cells including breast and CRC where the inhibition mechanism is involved with the binding of free β-catenin in the cytoplasm and the β-catenin–transcription factor 4 complex in the nucleus at the β-catenin binding pocket with amino acid residues, Lys345, Trp383, Arg386 and Asn415. Similarly, the inhibitions of MDM2 protein and KRAS were controlled by the binding of inhibitor toward the binding pocket. The p53 binding pocket on MDM2 protein was selected where it controls the activity of p53 protein through working as ubiquitin E3 ligase promoting p53 degradation through the proteasome degradation pathway [33] whereas the binding of ligand on a defined KRAS active site will regulate the cell growth, differentiation, and apoptosis through several signalling pathways [34].

Among the eight isolated anthraquinones tested, morindone showed the highest inhibition against multiple targets in CRC which include β-catenin, MDM2-p53, and KRAS as reported in Table 1. Another interesting compound having comparable binding affinity with morindone is rubiadin. Both docked complexes are selected for a binding mode analysis. The three-dimensional (3D) structure of morindone (cyan) and rubiadin (yellow) against β-catenin with the TCF cofactor shown in red, MDM2-p53, and KRAS where molecular surface regarding residues type in 3 Å binding with compounds were illustrated in Fig 2. Amino acid residue types are coloured by YASARA colouring scheme: nonpolar, grey; amidic, green; basic, blue; acidic, red; hydroxylic, cyan. A two-dimensional (2D) binding mode inspection using Biovia

**Table 1. Binding affinity of anthraquinone compounds 1–8 towards β-catenin, MDM2-p53 and KRAS in colorectal cancer.**

| Anthraquinone compound | Binding affinity (kcal/mol) | | | |
|---|---|---|---|---|
| | β-catenin | | MDM2-p53 | KRAS |
| | Site A | Site B | | |
| Nordamnacanthal (1) | **-6.6** | -4.8 | -6.9 | -7.6 |
| Damnacanthal (2) | -5.7 | -5.1 | -6.6 | -7.6 |
| 1,3,5-trihydoxy-2-methoxy-6-methyl anthraquinone (3) | -5.6 | **-5.2** | -6.8 | -7.2 |
| Morindone (4) | -5.9 | -4.9 | **-7.1** | **-8.5** |
| Sorendidiol (5) | -5.8 | -4.7 | **-7.1** | -8.2 |
| Rubiadin (6) | **-6.6** | -4.7 | **-7.1** | -7.7 |
| Damnacanthol (7) | -5.4 | -5.0 | -6.7 | -7.6 |
| Lucidin-ω-methylether (8) | -6.3 | -4.6 | -7.0 | -7.4 |

The best scores were highlighted in bold.

Discovery Studio (Fig 3) revealed the important binding interactions of morindone and rubiadin with residues at the binding site of protein receptors.

In the case of binding with β-catenin, all anthraquinones preferred to bind at hydrophobic pocket site A where the important amino acids in chain A are Gln,309, Lys312, Val349, Cys350, Ser351, Ser352 and residues in TCF cofactors in chain B are Glu29, Lys30. Lys312 was identified as the key amino acids in this binding pocket in several reports [29, 35]. Rubiadin bound different pattern in the same pocket with slightly more favourable interaction of -6.6kcal/mol compared to -5.9kcal/mol for morindone. Rubiadin exhibits better binding due to hydrogen bond interaction of carbonyl oxygen with Lys312 where oxygen atom of the chelated hydroxyl in morindone was found to have hydrogen bonding with Ser351, not with Lys312. Another interaction type, π-sulfur interaction with Cys350 was also shown in Fig 2.

Targeting MDM2-p53 interaction received great attention. Overexpression of MDM2 and subsequent deactivation of p53 protein resulted in failure of apoptosis and cancer cell survival [36–40]. In previous study, 14 amino acids form a deep hydrophobic cavity on the MDM2 protein structure that can be occupied by small molecule inhibitors: Leu54, Leu57, Ile61, Met62, Tyr67, Gln72, Val75, Phe86, Phe91, Val93, His96, Ile99, Tyr100, and Ile101 [33]. From our *in-silico* investigation, morindone and rubiadin have the same binding affinity of -7.1kcal/mol. The binding pocket includes nonpolar residues; Ile19, Gln24, Lys51, His96, Tyr100 and polar residues; Leu54, Val93, and Ile99 where the vdW and π-Alkyl interactions were dominated, respectively. Another type of interaction, π-π stacking was observed in rubiadin and His96.

KRAS are the most frequently mutated oncogenes in human cancer among the three human Ras genes (KRAS, NRAS and HRAS) appearing in 45% of CRC which makes Ras one of the most crucial targets in oncology for drug development [41]. Morindone exhibits the best scores among the tested anthraquinones with a binding affinity of -8.5kcal/mol toward KRAS where its interactions with β-catenin and MDM2-p53 were -5.9 and -7.1kcal/mol, respectively. Two oxygen atoms of the chelated hydroxyl formed H-bond interactions with Lys117. π-Alkyl interaction with Ala18, Tyr32, Ala146 and π-π stacked interaction with Phe28 of the planar benzyl ring in morindone have been observed. Besides, the nonpolar interaction in the hydrophobic pockets of Lys117, Asn116, Leu120, Ser145, Lys147, Thr148 was observed. On the other hand, different binding mode of interaction of rubiadin were due to two hydrogen bonding of carbonyl oxygen and oxygen atoms of the chelated hydroxyl with Ala18, Asn116; π-Anion interaction with Lys117; π-Alkyl interaction with Tyr32, Leu120; π-π stacked interaction with Phe28; and hydrophobic interaction with Gly15, Lys16, Ser17, Ala18, Val29, Asp30,

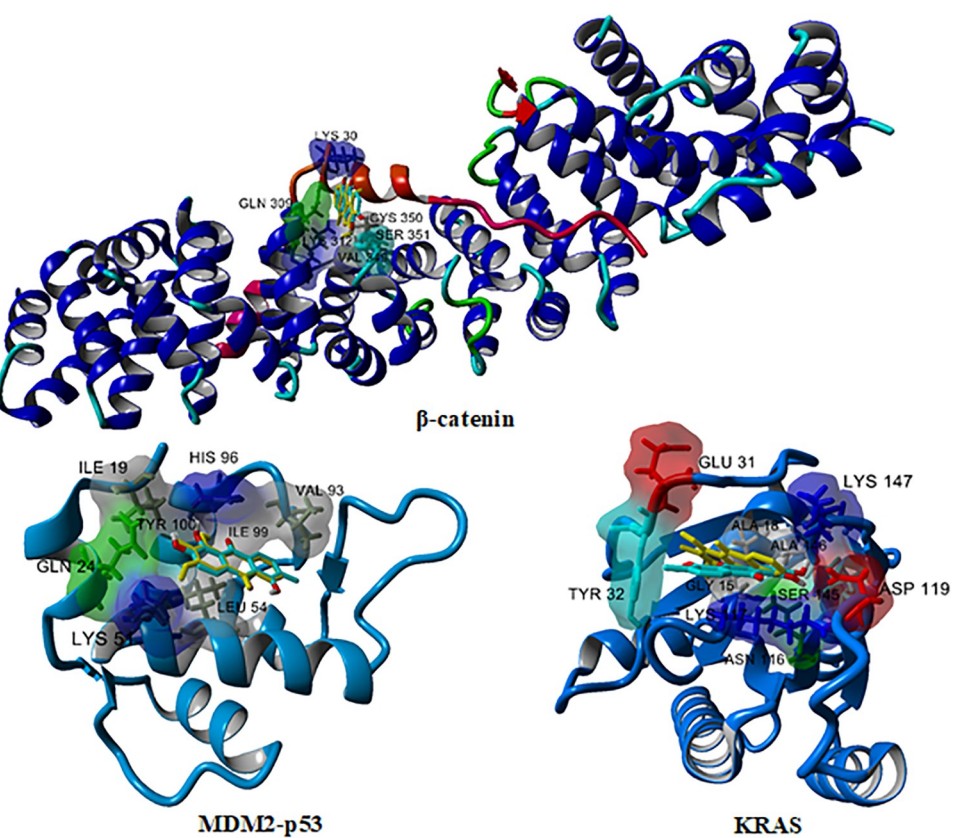

**Fig 2. Docking complex of morindone (cyan) and rubiadin (yellow) against β-catenin, MDM2-p53, and KRAS.**
The TCF cofactor in β-catenin was shown in red. Molecular surface regarding residues type at the binding residues within 3 Å from compounds were illustrated. Amino acid residue types were colored by YASARA coloring scheme: nonpolar, grey; amidic, green; basic, blue; acidic, red; hydroxylic, cyan.

Glu31, Asp33, Ser145, Asn116. A similar trend was demonstrated by rubiadin that bound better with KRAS, MDM2, and β-catenin at a binding affinity of -7.7, -7.1, and -6.6kcal/mol, respectively.

## Cytotoxic activity evaluation

All compounds were evaluated with cytotoxicity test using MTT assay on normal colon, CCD841 CoN cells and three CRC cell lines, HCT116 cells, LS174T cells and HT29 cells. The $IC_{50}$ values are presented in Table 2. The cytotoxicity effect of eight anthraquinone compounds, 5-Fluouracil (5-FU) and doxorubicin hydrochloride (DOX) on the cell viability of each cell lines was illustrated in Figs 4–7. Selectivity index (SI) which indicates the cytotoxicity selectivity was also calculated using $IC_{50}$ value of normal colon (CCD841 CoN cells) against $IC_{50}$ value of colorectal cancer cell lines and presented in Table 3. The bigger the SI value, the more selective it is. An SI value lower than 2 indicate general toxicity of the pure compound [42].

Generally, damnacanthal, 1,3,5-trihydoxy-2-methoxy-6-methyl anthraquinone, morindone, damnacanthol and lucidin-ω-methylether has $IC_{50}$ value lower than 25μM on HCT116 cells. Damnacanthal, morindone and sorendidiol has slightly higher $IC_{50}$ value ranged between 19μM and 30μM on HT29 cells. Only morindone has cytotoxic effect on LS174T cells at $IC_{50}$ value of 20.45 ± 0.03μM while nordamnacanthal and rubiadin has $IC_{50}$ value greater than

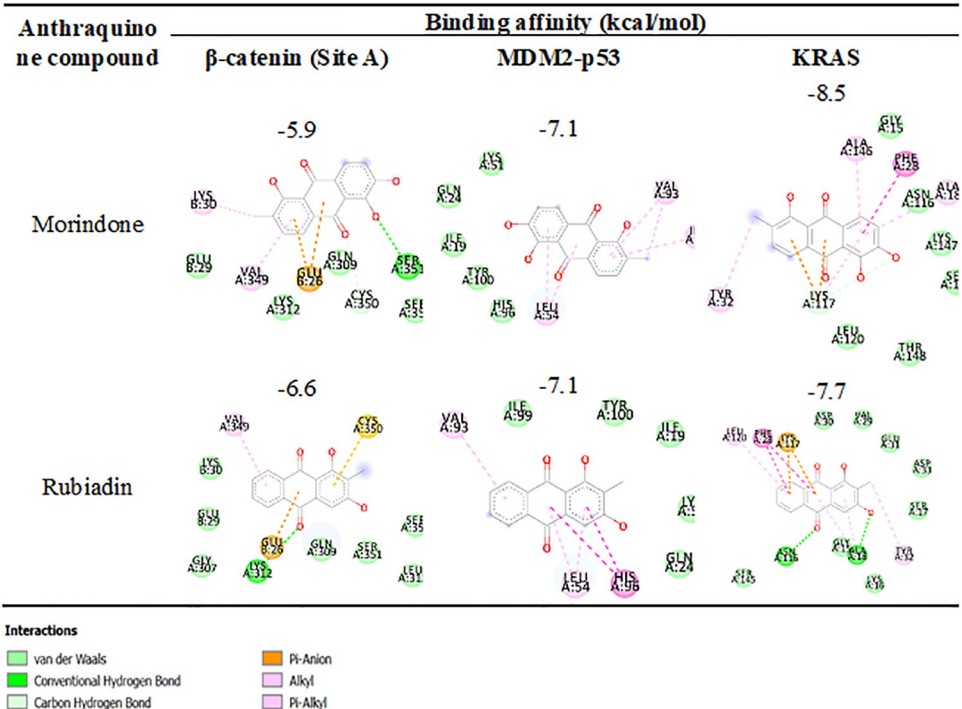

**Fig 3. Two-dimensional diagram of residues binding interaction of morindone and rubiadin with β-catenin (Site A), MDM2-p53, and KRAS.** Each residue is colored by the type of interaction; van der Waals (light green), hydrogen bond (green), and polar (magenta and orange).

50μM for all cell lines. Overall, all anthraquinone compounds have no significant cytotoxicity effect on the normal colon cells, CCD841 CoN cells.

An IC$_{50}$ value as low as 0.74 ± 0.06μM was exhibited by damnacanthal towards HCT116 cells, which is comparable with IC$_{50}$ value of 5-FU and DOX. Data shows that damnacanthal has relatively high SI value, 951.60 towards HCT116 cells and 24.85 towards HT29 cells. Good anticancer and antitumor activities by damnacanthal towards MCF-7 cells (breast cancer),

**Table 2. IC$_{50}$ value of anthraquinone compounds 1–8, 5-Fluorouracil, doxorubicin hydrochloride in three colorectal cancer cell lines and normal colon cells.**

| Anthraquinone compound | IC$_{50}$ (μM) | | | |
|---|---|---|---|---|
| | **HCT116** | **LS174T** | **HT29** | **CCD841 CoN** |
| Nordamnacanthal (1) | >50 | >50 | >50 | >50 |
| Damnacanthal (2) | **0.74 ± 0.06** * | >50 | **28.17 ± 0.08** | >50 |
| 1,3,5-trihydoxy-2-methoxy-6-methyl anthraquinone (3) | 24.57 ± 0.07 * | >50 | >50 | >50 |
| Morindone (4) | **10.70 ± 0.04** | **20.45 ± 0.03** | **19.20 ± 0.05** * | >50 |
| Sorendidiol (5) | >50 | >50 | 27.17 ± 0.07 | >50 |
| Rubiadin (6) | >50 | >50 | >50 | >50 |
| Damnacanthol (7) | 18.47 ± 0.02 | >50 | >50 | >50 |
| Lucidin-ω-methylether (8) | 14.88 ± 0.01 | >50 | >50 | >50 |
| Doxorubicin hydrochloride | 0.17 ± 0.04 | 2.27 ± 0.04 | 0.3 ± 0.06 | - |
| 5-Fluorouracil | 5.0 ± 0.04 | 4.16 ± 0.02 | 0.2 ± 0.05 | - |

Best IC$_{50}$ values were highlighted in bold. Data was shown as mean ± SD from three independent experiments. One * indicates p value <0.05 and considered significant

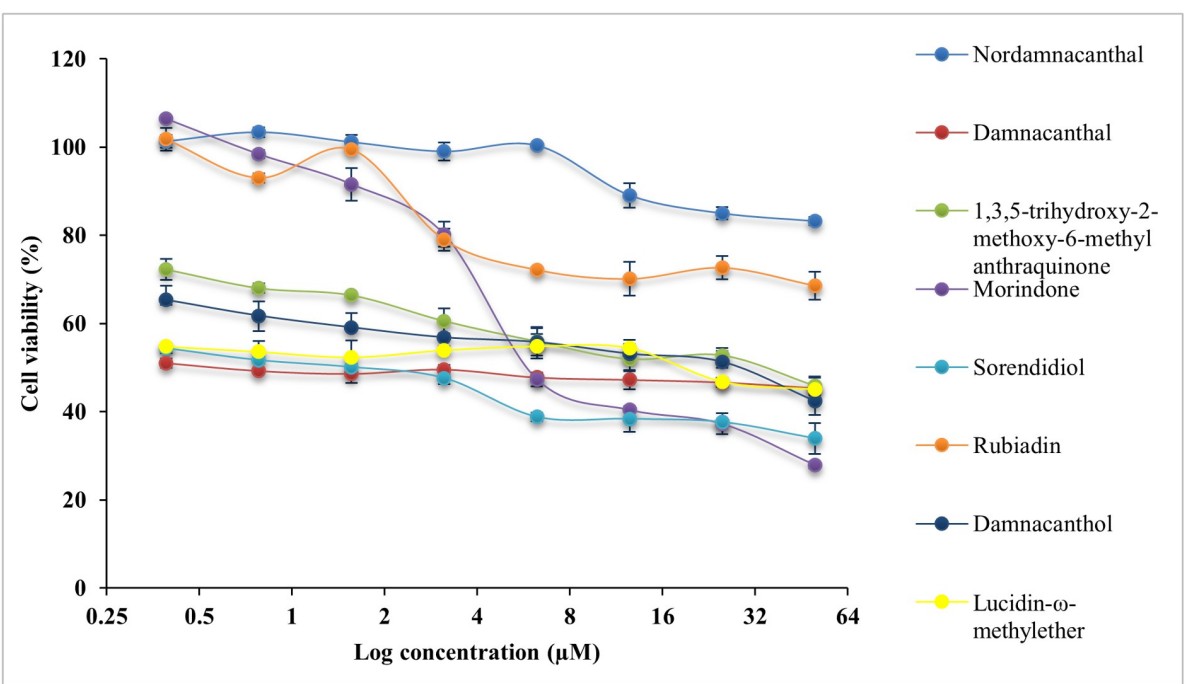

**Fig 4. Cytotoxicity effect of anthraquinone compounds 1–8 on the cell viability of HCT116 cells.** Data was shown as mean ± SD from three independent experiments.

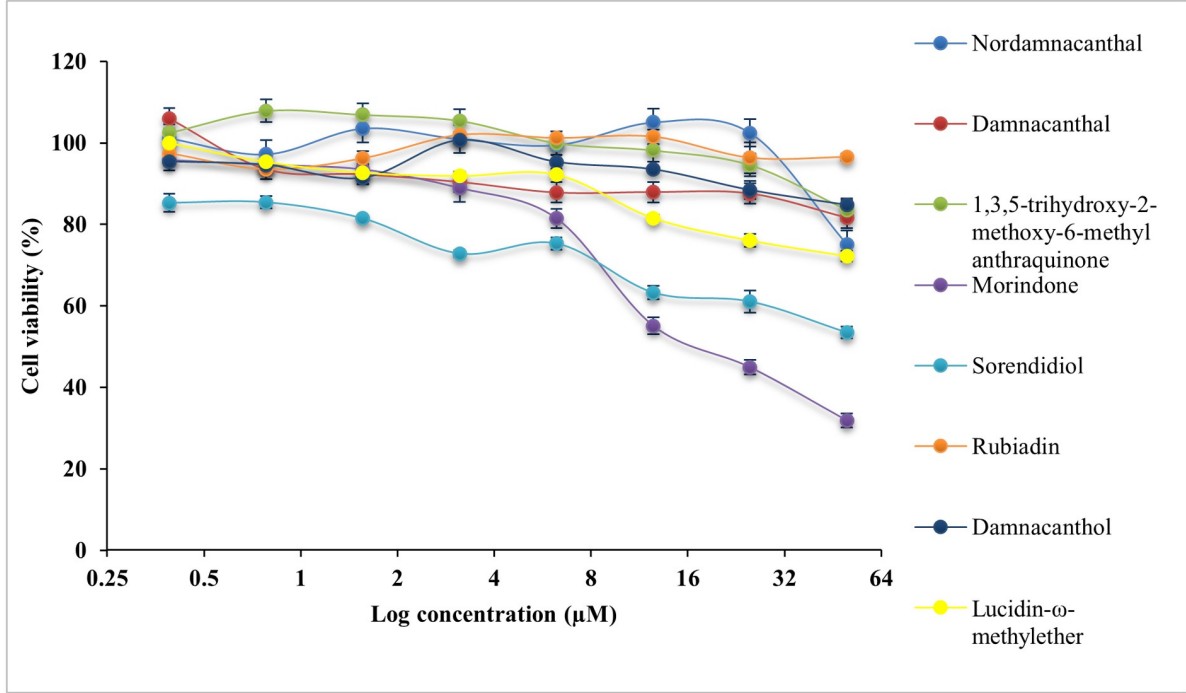

**Fig 5. Cytotoxicity effect of anthraquinone compounds 1–8 on the cell viability of LS174T cells.** Data was shown as mean ± SD from three independent experiments.

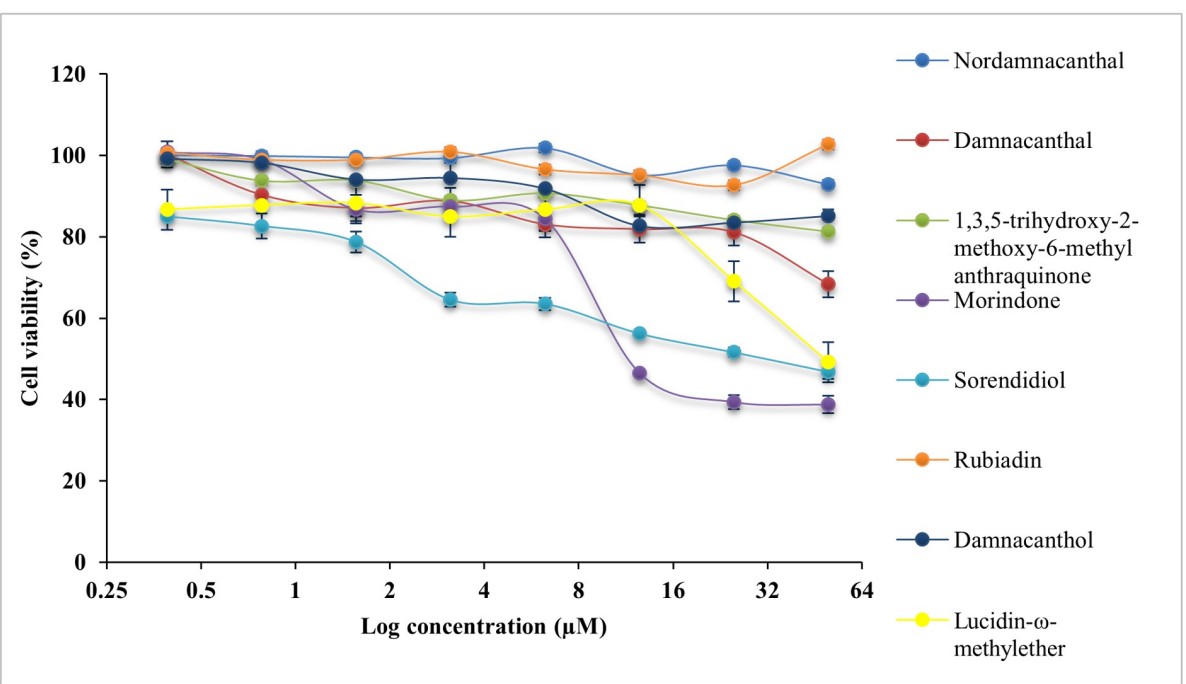

**Fig 6. Cytotoxicity effect of anthraquinone compounds 1–8 on the cell viability of HT29 cells.** Data was shown as mean ± SD from three independent experiments.

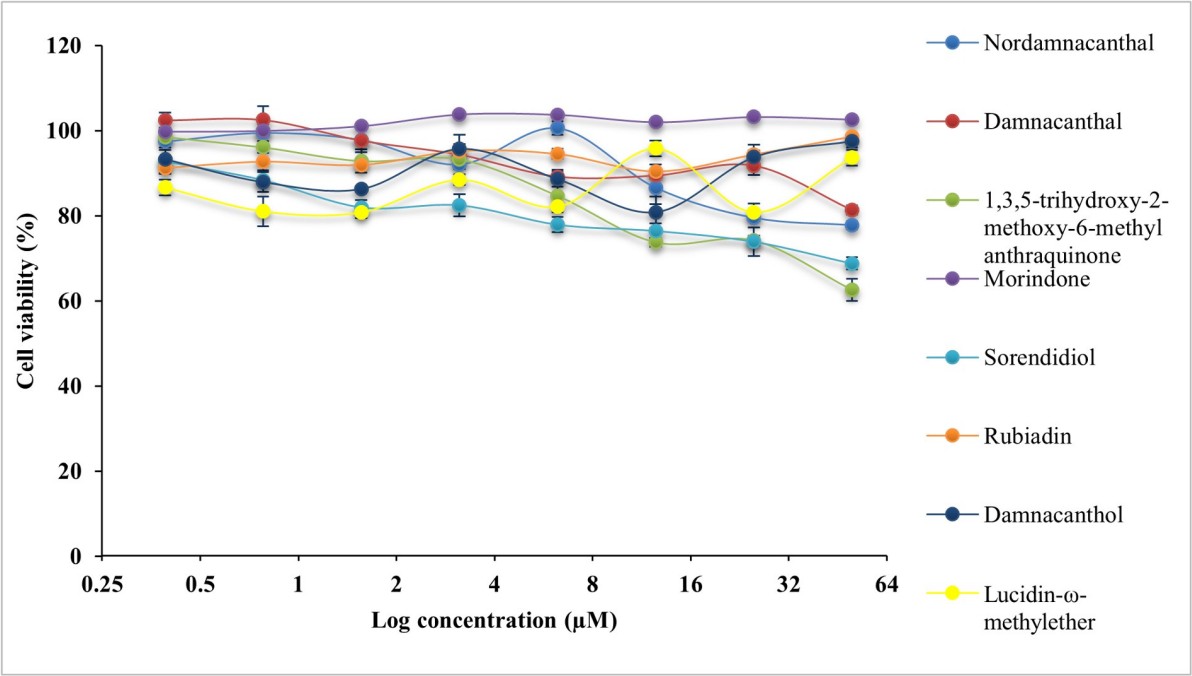

**Fig 7. Cytotoxicity effect of anthraquinone compounds 1–8 on the cell viability of CCD841 CoN cells.** Data was shown as mean ± SD from three independent experiments.

**Table 3. Selectivity index (SI) of normal colon cells against colorectal cancer cells.**

| Anthraquinone compound | HCT116 | LS174T | HT29 |
|---|---|---|---|
| Nordamnacanthal (1) | 0.97 | 4.79 | 0.42 |
| Damnacanthal (2) | **951.60** | 0.28 | **24.85** |
| 1,3,5-trihydoxy-2-methoxy-6-methyl anthraquinone (3) | 4.55 | 0.97 | 0.03 |
| Morindone (4) | **76.25** | **39.89** | **42.49** |
| Sorendidiol (5) | 7.10 | <0 | **29.19** |
| Rubiadin (6) | <0 | <0 | <0 |
| Damnacanthol (7) | <0 | 0.59 | 1.46 |
| Lucidin-ω-methylether (8) | <0 | <0 | <0 |

High SI values were highlighted in bold.

HCT116 cells (colorectal cancer) and HepG2 cells (hepatocellular carcinoma) were also reported by [43–45]. A dose-dependent manner of antiproliferative activity of damnacanthal treated cancer cells was shown by [44] which corresponds with this study. The strong cytotoxicity activity of damnacanthal is highly linked to the presence of a formyl group at Carbon 1 and hydroxylated at Carbon 3 [46]. Despite of a moderate binding affinity on three protein targets was exhibited by damnacanthal, this compound has displayed significant cytotoxic effect and selectivity index towards HCT116 cells, which are comparable to the standard drugs, 5-FU, and DOX. Therefore, damnacanthal could be a promising candidate in targeting KRAS mutation in CRC as HCT116 cells carry KRAS gene mutation.

Meanwhile, morindone also has high cytotoxicity effect towards all three CRC cell lines. The $IC_{50}$ value of morindone for HCT116 cells was $10.70 \pm 0.04\mu M$, $20.45 \pm 0.03\mu M$ for LS174T cells and $19.20 \pm 0.05\mu M$ for HT29 cells. Morindone was selective towards all three CRC cell lines with SI value of 76.25 for HCT116 cells, 39.89 for LS174T cells and 42.49 for HT29 cells. This was in concordance with the molecular docking results obtained. Among eight anthraquinones, only morindone demonstrated strongest binding affinity towards both MDM2-p53 and KRAS. This indicates that morindone is favourable for targeting the p53 and KRAS mutation in CRC. The high degree of cytotoxicity effect performed by morindone could be due to the formyl group appear at 1,2-dihydroxyl group [46]. The inhibitory effect of morindone on HCT116 cell proliferation was also reported in work by [23]. Other than that, morindone also demonstrated strong antimicrobial activity against *C. lipolytica* [46] and oxacillin-resistant *Staphylococcus aureus* [47].

## Drug combination assay

Both 5-FU and DOX have limited therapeutic efficacy in CRC, owing to the presence of drug resistance and associated side effects [48, 49]. The incorporation of phytochemical compounds to anticancer drug in cancer prevention and treatment brings promising outcomes and improves survival rate [50], yet insufficient data presented on anthraquinone compounds. Therefore, drug combination assay was carried out to assess whether an enhanced potency against CRC cells can be achieved by anthraquinone compounds and anticancer drugs like DOX and 5-FU. Only damnacanthal and morindone with optimal $IC_{50}$ values were selected to test in HCT116 cells and HT29 cells, which these two compounds established selective cytotoxicity towards. The $IC_{50}$ value in combination and combination index was presented in Table 4. The results showed that for the combination of damnacanthal and DOX, the $IC_{50}$ value has dropped to $0.12 \pm 0.05\mu M$ in HCT116 cells and $0.33 \pm 0.08\mu M$ in HT29 cells. A lowered concentration dose ($CD_{50}$) value of damnacanthal and DOX in breast cancer cells, MCF-7 was

**Table 4. IC$_{50}$ and combination index (CI) values.**

| Combination pair | IC$_{50}$ (μM) | | Combination index | |
|---|---|---|---|---|
| | **HCT116** | **HT29** | **HCT116** | **HT29** |
| Damnacanthal + Doxorubicin hydrochloride | 0.12 ± 0.05 | 0.33 ± 0.08 | **0.87** | 1.11 |
| Damnacanthal + 5-Fluorouracil | 0.49 ± 0.07 * | 0.17 ± 0.06 | **0.83** | **0.86** |
| Morindone + Doxorubicin hydrochloride | 0.05 ± 0.04 | 0.19 ± 0.05 * | **0.36** | **0.88** |
| Morindone + 5-Fluourouracil | 2.08 ± 0.04 * | 0.14 ± 0.09 | **0.69** | **0.71** |

Table showed the IC$_{50}$ and combination index (CI) values for combination treatment of anthraquinone compounds and drugs in HCT116 and HT29 cells. Data was shown as mean ± SD from three independent experiments. CI value <1 = synergism was highlighted in bold. One * indicates p value <0.05 and considered significant.

also reported by [51]. Likewise, the IC$_{50}$ value of damnacanthal and 5-FU combination in HCT116 cells and HT29 cells has decreased to 0.49 ± 0.07μM and 0.17 ± 0.06μM too. As for the combination of 5-FU and damnacanthal, the cell growth was inhibited by reduced IC$_{50}$ value of 0.05 ± 0.04μM in HCT116 cells and 0.21 ± 0.1μM in HT29 cells. Identical mark down on IC$_{50}$ value to 2.08 ± 0.04μM in HCT116 cells and 0.14 ± 0.09μM in HT29 cells was observed in the combination of 5-FU and morindone. With that, the combination index (CI) was calculated to figure whether the effect is synergistic (greater), addictive (neutral) or antagonistic (reduced) [50]. Synergistic interaction was observed in majority pairs of combination, with the most prominent CI at 0.36 displayed by the pair of morindone and DOX in HCT116 cells, indicating high synergism. Amongst all, only the combination of damnacanthal and DOX exhibited antagonistic effect with a CI value of 1.11 despite a reduce in IC$_{50}$ value. This result suggested the addition of morindone to anticancer drugs aside from 5-FU and DOX may elevate the therapeutic efficacy and deplete the side effect, given the much-decreased IC$_{50}$ value. Nevertheless, other pairs that demonstrates synergism effect also should be further studied for their anticancer properties.

## Characterization of morindone

As the most active anthraquinone compound in this study, morindone (4) was characterized by Fourier-transform infrared (FTIR) spectroscopy (Zamakshshari et al., 2017), gas chromatography-mass spectrometry (GC-MS), and NMR (Fig 8). The $^1$H NMR (400 MHz) and $^{13}$C NMR (125 MHz) spectral data of morindone in Acetone-d$_6$ are available in Supplementary information.

## Conclusion

Generally, phytochemical compound offers great anticancer properties in cancer treatment by interfering carcinogenesis and modulating signaling pathway. Our study concluded that among the 8 anthraquinones, morindone showed high cytotoxicity effect and great selectivity index towards colorectal cancer cell lines as well as strong binding affinities in *in-vitro* and *in-silico* investigation towards multiple protein targets of β-catenin, MDM2-p53, and KRAS in comparison to other anthraquinones. Molecular docking result showed that all the active compounds can bind well into each targeted protein receptor by binding to important residues in the protein, explaining their inhibitory activities. Interesting binding activity of selected anthraquinones were contributed by the presence of several hydrogen bonds from carbonyl oxygen, oxygens at chelate hydroxyl, Van der Waals forces, and π-interactions of the planar benzyl ring. Further *in vivo* assays towards morindone are needed for the development of potential lead compound against colorectal cancer cells.

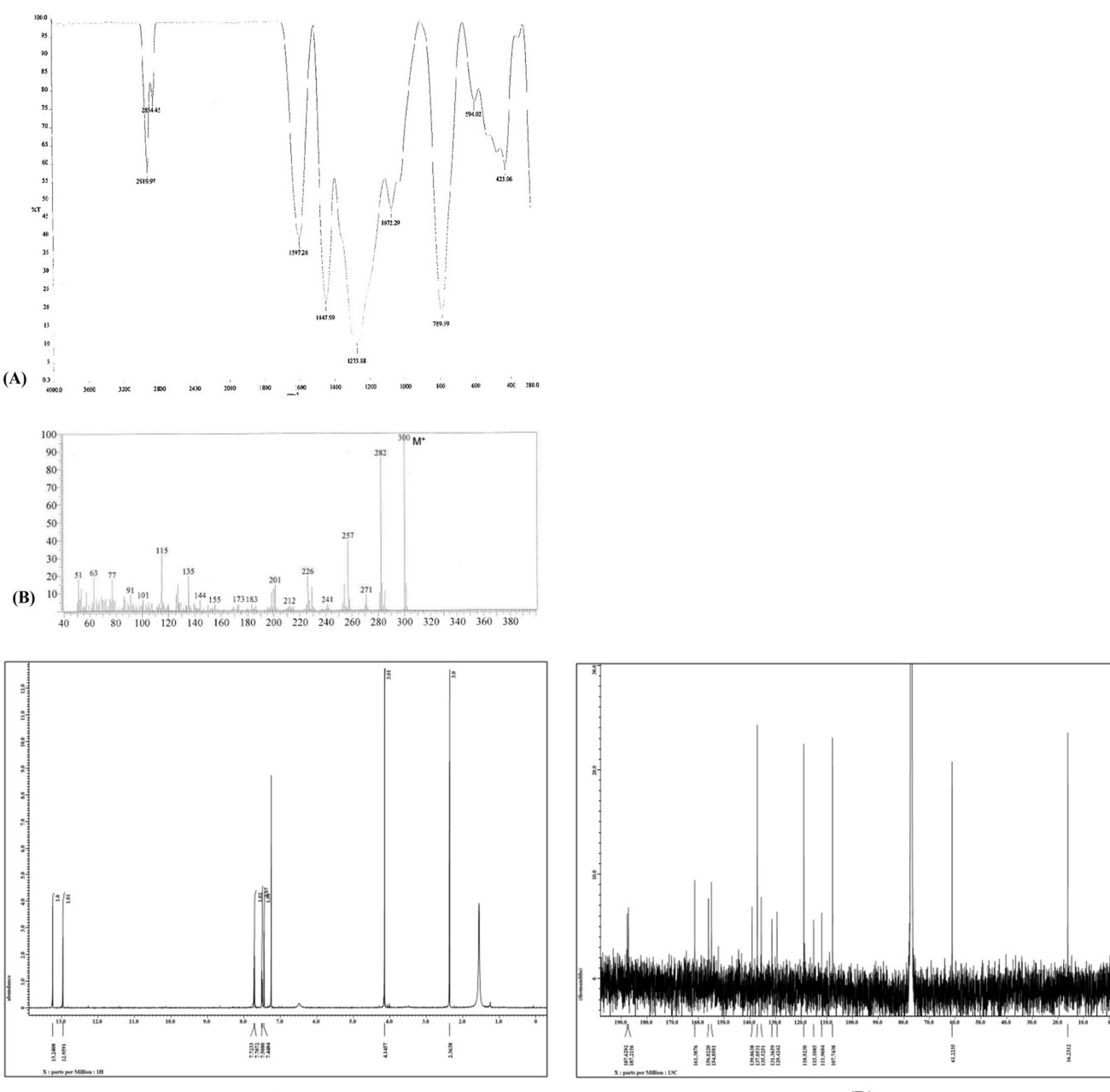

**Fig 8. Characterization of morindone.** (A) FTIR spectrum of morindone. (B) GC-MS chromatogram of morindone. (C) $^1$H NMR spectra of morindone in Acetone-d$_6$. (D) $^{13}$C NMR spectra of morindone in Acetone-d$_6$.

## Supporting information

**S1 File.**
(PDF)

**S1 Table. $^1$H NMR (400 MHz) and $^{13}$C NMR (125 MHz) spectral data of morindone (4) in Acetone-d$_6$.** δ in ppm, J in Hz.
(PDF)

## Acknowledgments

Authors would like to thank Dr Chin Fei Chee for providing 5-fluouracil and Doxorubicin hydrochloride.

## Author Contributions

**Conceptualization:** Rozana Othman, Najihah Mohd Hashim.

**Data curation:** Yean Kee Lee, Nurshamimi Nor Rashid.

**Funding acquisition:** Iskandar Abdullah, Nurshamimi Nor Rashid.

**Investigation:** Cheok Wui Chee, Nurshamimi Nor Rashid.

**Methodology:** Cheok Wui Chee, Nor Hisam Zamakshshari, Nurshamimi Nor Rashid.

**Resources:** Rozana Othman.

**Software:** Cheok Wui Chee, Vannajan Sanghiran Lee.

**Supervision:** Vannajan Sanghiran Lee, Najihah Mohd Hashim, Nurshamimi Nor Rashid.

**Validation:** Cheok Wui Chee, Vannajan Sanghiran Lee, Najihah Mohd Hashim, Nurshamimi Nor Rashid.

**Writing – original draft:** Cheok Wui Chee, Vannajan Sanghiran Lee.

**Writing – review & editing:** Najihah Mohd Hashim, Nurshamimi Nor Rashid.

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
