## [Decision Letter · Decision Letter 0]

14 Feb 2022

PONE-D-21-32928Morindone from Morinda citrifolia as a potential antiproliferative agent against colorectal cancer cell lines.PLOS ONE

Dear Dr. Nor Rashid,

Thank you for submitting your manuscript to PLOS ONE. After careful consideration, we feel that it has merit but does not fully meet PLOS ONE’s publication criteria as it currently stands. Therefore, we invite you to submit a revised version of the manuscript that addresses the points raised during the review process.

We look forward to receiving your revised manuscript.

Kind regards,

Chakrabhavi Dhananjaya Mohan, Ph.D

Academic Editor

PLOS ONE

Journal Requirements:

2. ‘Please include your tables as part of your main manuscript and remove the individual files. Please note that supplementary tables (should remain/ be uploaded) as separate "supporting information" files

4. PLOS requires an ORCID iD for the corresponding author in Editorial Manager on papers submitted after December 6th, 2016. Please ensure that you have an ORCID iD and that it is validated in Editorial Manager. To do this, go to ‘Update my Information’ (in the upper left-hand corner of the main menu), and click on the Fetch/Validate link next to the ORCID field. This will take you to the ORCID site and allow you to create a new iD or authenticate a pre-existing iD in Editorial Manager. Please see the following video for instructions on linking an ORCID iD to your Editorial Manager account: https://www.youtube.com/watch?v=_xcclfuvtxQ.

5. Please amend the manuscript submission data (via Edit Submission) to include author “Chin Fei Chee”.

“The authors would like to acknowledge financial support from Universiti Malaya under IIRG-003A-2019, IIRG-003B-2019 and IIRG-003C-2019 research grants and research facilities.”

“NNR, IIRG003C-2019

CCF, IIRG003B-2019

IA, IIRG003A-2019

Impact-Oriented Interdisciplinary Research  Grant Programme

The funders had no role in study design, data collection and analysis, decision”

7. Please ensure that you refer to Figure 4, 5 and 6 in your text as, if accepted, production will need this reference to link the reader to the figure.

8. We note that you have stated that you will provide repository information for your data at acceptance. Should your manuscript be accepted for publication, we will hold it until you provide the relevant accession numbers or DOIs necessary to access your data. If you wish to make changes to your Data Availability statement, please describe these changes in your cover letter and we will update your Data Availability statement to reflect the information you provide

Reviewers' comments:

Reviewer's Responses to Questions

**Comments to the Author**

1. Is the manuscript technically sound, and do the data support the conclusions?

Reviewer #1: Yes

Reviewer #2: Yes

2. Has the statistical analysis been performed appropriately and rigorously? 

Reviewer #1: Yes

Reviewer #2: Yes

3. Have the authors made all data underlying the findings in their manuscript fully available?

Reviewer #1: Yes

Reviewer #2: Yes

4. Is the manuscript presented in an intelligible fashion and written in standard English?

Reviewer #1: Yes

Reviewer #2: Yes

5. Review Comments to the Author

Reviewer #1: Search for new phytochemicals, with anti-cancer properties is an ever green area of research in the field of drug discovery, In view of this the authors in the present manuscript entitled “Morindone from Morinda citrifolia as a potential antiproliferative agent against colorectal cancer cell lines” have studied the Anti-cancer potentials of root barks of Morinda citrifolia and isolated the eight anthraquinone derivatives.

Even though the anticancer properties of anthraquinone compounds from Morinda citrifolia is previously studied in this present work, the authors have identified the key targets of these active compounds by protein docking studies. They have identified the binding sites for these compounds in the three important genes including P53, KRAS and β- catenin which were routinely mutated in colorecral cancer (CRC).

Comment #1.

In the Cytotoxic assay carried out the authors have used three human colorectal cancer cell lines: HT29 (with p53 and APC mutation), LS174T (with KRAS mutation) and HCT116 (with KRAS mutation). In this study HCT116 wells were seeded at 7500 cells/well while other cells were seeded at 30,000 cells/well. What is the reason for varying the number of cells seeded in the other cell lines?

Comment #2.

In the Introduction part in the last paragraph one of the reference needs to be corrected. It is written as (et al., 2016), without the author’s name. mention the author’s name as well.

Comment #3.

In the figures 3,4,5,6 representing the Cytotoxicity effect of anthraquinone compounds 1-8 in different cancer cell lines and normal cells, the graph could be represented in a colored line, which makes the lines distinguishable clearly.

The authors have identified that Morindone has high cytotoxicity effect and great selectivity index towards colorectal cancer cell lines as well as strong binding affinities in in-vitro and in-silico investigation towards multiple protein targets of β-catenin, MDM2-p53, and KRAS in comparison to other anthraquinones. In detail Molecular docking studies have shown that all the active compounds can bind well into each targeted protein receptor by binding to important residues in the protein, explaining their inhibitory activities. Further in vivo assays with Morindone are needed for the development of potential lead compound against colorectal cancer cells.

Overall, the manuscript could be accepted for publication once the above comments are considered and thoroughly revised again for any topographical error.

Reviewer #2: Review comments

Manuscript Number: PONE-D-21-32928

Dr. Nurshamimi and coworkers describes the Morindone from Morinda citrifolia as a potential antiproliferative agent against colorectal cancer cell lines.

The work carried out by this group is extension work of Kamiya group but this group used three colorectal cancer cell lines and activity shown by all the three cell lines was good.

After isolation and in vitro studies, binding efficiency of all the isolated derivatives were investigated. From the studies it is confirmed that Morindone is showed moderate inhibitory activity against multitargets like β-catenin, MDM2-p53 and KRAS.

1) Provided IR stretching frequencies isolated anthraquinone derivatives is not clear. All the derivatives with hydroxyl group but authors not mentioned the stretching frequencies of in data.

2) Include 1H NMR and 13C NMR data of active molecule Morindone in the manuscript.

3) Include in vivo assay results.

After all these modification paper will be considered.

I am suggesting major revision.

6. PLOS authors have the option to publish the peer review history of their article (what does this mean?). If published, this will include your full peer review and any attached files.

Reviewer #1: No

Reviewer #2: No

---

## [Author Response · Author response to Decision Letter 0]

24 Mar 2022

Response to reviewers

Title: Morindone from morinda citrifolia as a potential antiproliferative agent against colorectal cancer cell lines

Reference number: PONE-D-21-32928

Reviewer #1: Search for new phytochemicals, with anti-cancer properties is an ever green area of research in the field of drug discovery, In view of this the authors in the present manuscript entitled “Morindone from Morinda citrifolia as a potential antiproliferative agent against colorectal cancer cell lines” have studied the Anti-cancer potentials of root barks of Morinda citrifolia and isolated the eight anthraquinone derivatives.

Even though the anticancer properties of anthraquinone compounds from Morinda citrifolia is previously studied in this present work, the authors have identified the key targets of these active compounds by protein docking studies. They have identified the binding sites for these compounds in the three important genes including P53, KRAS and β- catenin which were routinely mutated in colorecral cancer (CRC).

Comment #1.

In the Cytotoxic assay carried out the authors have used three human colorectal cancer cell lines: HT29 (with p53 and APC mutation), LS174T (with KRAS mutation) and HCT116 (with KRAS mutation). In this study HCT116 wells were seeded at 7500 cells/well while other cells were seeded at 30,000 cells/well. What is the reason for varying the number of cells seeded in the other cell lines?

Response: The cell seeding density for each cell line has been optimized according to their proliferation rate and confluency at the end of treatment, hence the differences. HCT 116 cells showed faster proliferation rate while the other cells showed slower proliferation rate.

Comment #2.

In the Introduction part in the last paragraph one of the reference needs to be corrected. It is written as (et al., 2016), without the author’s name. mention the author’s name as well.

Response: Thank you for the comment. This has been updated with Vancouver style referencing, which is the PloS One reference style. The changes can be tracked in the Introduction section in the revised manuscript.

Comment #3.

In the figures 3,4,5,6 representing the Cytotoxicity effect of anthraquinone compounds 1-8 in different cancer cell lines and normal cells, the graph could be represented in a colored line, which makes the lines distinguishable clearly.

Response: Thank you for the suggestion. We have changed them to the colored line graph in Figures 4 – 7.

The authors have identified that Morindone has high cytotoxicity effect and great selectivity index towards colorectal cancer cell lines as well as strong binding affinities in in-vitro and in-silico investigation towards multiple protein targets of β-catenin, MDM2-p53, and KRAS in comparison to other anthraquinones. In detail Molecular docking studies have shown that all the active compounds can bind well into each targeted protein receptor by binding to important residues in the protein, explaining their inhibitory activities. Further in vivo assays with Morindone are needed for the development of potential lead compound against colorectal cancer cells.

Overall, the manuscript could be accepted for publication once the above comments are considered and thoroughly revised again for any topographical error.

Reviewer #2: Review comments

Manuscript Number: PONE-D-21-32928

Dr. Nurshamimi and coworkers describes the Morindone from Morinda citrifolia as a potential antiproliferative agent against colorectal cancer cell lines.

The work carried out by this group is extension work of Kamiya group but this group used three colorectal cancer cell lines and activity shown by all the three cell lines was good.

After isolation and in vitro studies, binding efficiency of all the isolated derivatives were investigated. From the studies it is confirmed that Morindone is showed moderate inhibitory activity against multitargets like β-catenin, MDM2-p53 and KRAS.

1) Provided IR stretching frequencies isolated anthraquinone derivatives is not clear. All the derivatives with hydroxyl group but authors not mentioned the stretching frequencies of in data.

Response: The IR stretching frequencies data of all 8 anthraquinone compounds has been published in Zamakshshari et al., 2017 – Cytotoxic activities of anthraquinones from morinda citrifolia towards SNU-1 and LS-174-T and K562 cell lines. Since this work is a continuation from Zamakshshari et al., 2017, we would think that there is no need to provide it again.

2) Include 1H NMR and 13C NMR data of active molecule Morindone in the manuscript.

Response: The data is provided in the supporting information file named S9_Table.pdf. This change can be tracked in the Supporting Information section in the revised manuscript. 

3) Include in vivo assay results.

Response: We are looking forward to test this morindone using animal model. However, due to the limitation of the current funding, we are not able to test the morindone using animal model. Therefore, the in vivo work will be our next step, once we receive the upcoming funding for the study.

After all these modification paper will be considered.

I am suggesting major revision.

---

## [Decision Letter · Decision Letter 1]

2 Jun 2022

PONE-D-21-32928R1Morindone from Morinda citrifolia as a potential antiproliferative agent against colorectal cancer cell lines.PLOS ONE

Dear Dr. Nor Rashid,

Thank you for submitting your manuscript to PLOS ONE. After careful consideration, we feel that it has merit but does not fully meet PLOS ONE’s publication criteria as it currently stands. Therefore, we invite you to submit a revised version of the manuscript that addresses the points raised during the review process.

We look forward to receiving your revised manuscript.

Kind regards,

Chakrabhavi Dhananjaya Mohan, Ph.D

Academic Editor

PLOS ONE

Journal Requirements:

Reviewers' comments:

Reviewer's Responses to Questions

**Comments to the Author**

1. If the authors have adequately addressed your comments raised in a previous round of review and you feel that this manuscript is now acceptable for publication, you may indicate that here to bypass the “Comments to the Author” section, enter your conflict of interest statement in the “Confidential to Editor” section, and submit your "Accept" recommendation.

Reviewer #1: All comments have been addressed

Reviewer #2: (No Response)

2. Is the manuscript technically sound, and do the data support the conclusions?

Reviewer #1: Yes

Reviewer #2: Partly

3. Has the statistical analysis been performed appropriately and rigorously? 

Reviewer #1: Yes

Reviewer #2: Yes

4. Have the authors made all data underlying the findings in their manuscript fully available?

Reviewer #1: Yes

Reviewer #2: No

5. Is the manuscript presented in an intelligible fashion and written in standard English?

Reviewer #1: Yes

Reviewer #2: Yes

6. Review Comments to the Author

Reviewer #1: The authors have considered the previously suggested corrections and included them in the revised manuscript. The manuscript can now be accepted without need of further revision

Reviewer #2: (No Response)

7. PLOS authors have the option to publish the peer review history of their article (what does this mean?). If published, this will include your full peer review and any attached files.

Reviewer #1: No

Reviewer #2: No

---

## [Author Response · Author response to Decision Letter 1]

15 Jun 2022

Response to reviewers

Title: Morindone from morinda citrifolia as a potential antiproliferative agent against colorectal cancer cell lines

Reference number: PONE-D-21-32928

Reviewer #2: In this study, the authors extracted the potential anthraquinone compounds from the root bark of Morinda citrifolia. Eight potential anthraquinone compounds were successfully isolated, purified and tested for their anticancer efficacy.

Despite the fact that this work is a repetition of previously published work, authors must provide characterization data as well as scanned images of the active molecule.

Comment #1.

I have suggested that, provided IR stretching frequencies some of the functional groups are missing, provide an active molecule scanned IR spectra.

Response: Thank you very much on your suggestion. We have now provided the active molecule (morindone) IR spectra under figure 8A of the results section.

Comment #2.

The authors provided data in the table, but I suggest you to include scanned images of H1 NMR and C13 NMR of active molecule.

Response: Thank you for the comment. We have now included the scanned images of H1 NMR and C13 NMR of the active molecule (morindone) under figure 8C and 8D, respectively. 

Comment #3.

After all these modifications, the paper will be considered.

Response: Thank you very much for your valuable comments and they certainly improved our paper for publication.

---

## [Editor Report · Decision Letter 2]

22 Jun 2022

Morindone from Morinda citrifolia as a potential antiproliferative agent against colorectal cancer cell lines.

PONE-D-21-32928R2

Dear Dr. Nor Rashid,

We’re pleased to inform you that your manuscript has been judged scientifically suitable for publication and will be formally accepted for publication once it meets all outstanding technical requirements.

Kind regards,

Chakrabhavi Dhananjaya Mohan, Ph.D

Academic Editor

PLOS ONE
---

## [Editor Report · Acceptance letter]

29 Jun 2022

PONE-D-21-32928R2 

Morindone from *Morinda citrifolia* as a potential antiproliferative agent against colorectal cancer cell lines 

Dear Dr. Nor Rashid:

I'm pleased to inform you that your manuscript has been deemed suitable for publication in PLOS ONE. Congratulations! Your manuscript is now with our production department. 

Kind regards, 

on behalf of

Dr. Chakrabhavi Dhananjaya Mohan 

Academic Editor

PLOS ONE